# Screening for High Risk of Sleep Apnea in an Ambulatory Care Setting in Saudi Arabia

**DOI:** 10.3390/ijerph16030459

**Published:** 2019-02-05

**Authors:** Adeel Nazir Ahmad, Geraldine McLeod, Nada Al Zahrani, Haneen Al Zahrani

**Affiliations:** 1Department of Family Medicine, King Faisal Specialist Hospital and Research Centre, Jeddah 40047, Saudi Arabia; nal-zahrani1@kfshrc.edu.sa (N.A.Z.); haneen788@gmail.com (H.A.Z.); 2Christchurch Health and Development Study, Department of Psychological Medicine, University of Otago, Christchurch 8011, New Zealand; geri.mcleod@otago.ac.nz

**Keywords:** obstructive sleep apnea, Berlin Questionnaire, Epworth Sleepiness Scale adult, Saudi Arabia, cross-sectional

## Abstract

Sleep apnea is a potentially serious but under-diagnosed sleep disorder. Saudi Arabia has a high prevalence of hypertension, diabetes, obesity, and smoking, which are all major risk factors for sleep apnea. However, few studies report screening for sleep apnea in Saudi Arabia. A three-month prospective, questionnaire-based study, using the Berlin Questionnaire (BQ) and the Epworth Sleepiness Scale (ESS), screened 319 patients attending a family medicine clinic in Saudi Arabia for risk of sleep apnea. The results showed that when using the BQ and the ESS, 95 (29.8%) and 102 (32.0%) respondents were at high risk of sleep apnea. Taken together, the BQ and the ESS combined measure showed that 41 (12.9%) respondents were classified as high risk for sleep apnea. Logistic regression revealed that the high risk of sleep apnea was statistically significantly (*p* < 0.05) associated with respondent characteristics of obesity and hypertension. No associations were found between high risk for sleep apnea and: Smoking, diabetes mellitus, hypothyroidism or hyperlipidemia. Screening for sleep apnea using the BQ and ESS questionnaires, particularly among those who are obese or hypertensive, can be a fast, valid and acceptable way of alerting the physician to this disorder among patients.

## 1. Introduction

Sleep apnea is a serious medical condition characterized by the disruption of sleep due to intermittent pharyngeal collapses. As a result, hypoxia and ventilator overshoot hyperoxia lead to alterations in intrathoracic pressure, local airway edema and inflammation [1,2]. Risk factors for sleep apnea include: Being middle-aged or older, smoking, having a genetic predisposition for sleep-disordered breathing, certain medications, hypothyroidism and having a neurological disorder [2,3,4]. Sleep apnea is also associated with a number of comorbidities including cardiovascular problems such as hypertension and metabolic abnormalities such as obesity, hyperlipidemia and diabetes [2,5,6]. Sleep apnea can have adverse consequences such as cardiovascular mortality [7,8,9], stroke, cancer [2,8], depression [10] and reduced quality of life [2,11]. In addition to the adverse health and psychosocial effects of sleep apnea, reduced concentration and memory may be associated with driving and workplace accidents [2,12,13].

Due to sleep apnea’s negative consequences, the disorder has become a significant public health problem worldwide now affecting 5–15% of the general population [1,2,13,14,15,16]. The prevalence of sleep apnea in Saudi Arabia was initially assessed nearly a decade ago in middle-aged men and women [17,18]. BaHammam [17,18] showed that 33.3% of men and 39% of women were considered as being at a high risk for sleep apnea. 

Since that time, a systematic review by Abrishami [19] aimed to identify and evaluate available screening questionnaires of sleep apnea that could be used by physicians during consultations. The study showed that the most common screening tool was the Berlin Questionnaire (BQ) [20], which was used in four studies. The previous Saudi Arabian studies by BaHammam [17,18] also used the BQ to ascertain sleep apnea risk. The BQ is intended to predict sleep apnea from questions on snoring behavior, wake time sleepiness, obesity and hypertension [20]. However, there has been some disagreement among researchers regarding the utility of the BQ. Some authors consider the BQ to be a poor predictor of obstructive sleep apnea when used as the only screening tool [21] while others consider the BQ to be accurate if it is not used as a preoperative test [22]. The reasons for the disagreement on the accuracy of the BQ may stem from systematic differences between validation study patients and surgical patients [22].

The systematic review by Abrishami [19] also identified the Epworth Sleepiness Scale (ESS) as a potentially useful measure to detect daytime sleepiness among adults. The ESS has been shown to distinguish between controls and those with previously diagnosed sleep disorders, including sleep apnea [23]. Some concerns have been raised about the predictive value of the ESS as it may not distinguish simple snorers from patients with sleep apnea [19,22] or that it may have a high false negative rate [22]. However, for clinicians who are trying to identify those patients at high risk of sleep apnea, this measure could be supplemented with a patient interview or other clinical assessments such as polysomnography [22]. 

As the primary-care physician may most likely be the first to encounter the patient with sleep problems, it has been suggested that they should be responsible for sleep disorder recognition [24]. To assist with the identification of those at risk of sleep apnea, screening patients at their usual medical clinic may be a fast, valid and acceptable way of alerting the physician to the need to initiate further investigation [19,20,22]. As the initial sleep apnea research in Saudi Arabia was conducted nearly a decade ago, and since that time the prevalence of sleep apnea has increased across the world, it is important to update the information on the prevalence of sleep apnea risk among that population. Due to the need to correctly classify patients at high risk of sleep apnea, it is justifiable to use both the BQ and the ESS together as screening tools to assess the two key features of sleep apnea, namely snoring and excessive daytime sleepiness [21].

Against this background, this cross-sectional study aimed to use both the BQ and the ESS questionnaires to screen for sleep apnea among Saudi Arabian men and women from the age of 18 to 80 years attending a family medicine clinic in Jeddah. In addition to identifying patients at high risk of sleep apnea, we also aimed to determine the concurrent associations between the risk of sleep apnea and a series of potential predictors of sleep apnea of obesity, diabetes, hypothyroidism, hyperlipidemia and hypertension.

## 2. Materials and Methods 

Approximately 380 adult patients aged over 18 attending a family medicine clinic of the principal investigator over a three-month period (January to March 2015), were invited to participate. Of these, 61 refused. Patients were given an Arabic or English version of the questionnaires to complete before their appointment. The translational validity of the questionnaires was ensured by getting the questionnaires professionally translated from English to Arabic, then back translated from Arabic to English. The questionnaires were piloted by 10 bilingual physicians and patients for clinical validity. The questionnaires were:

Berlin Questionnaire (BQ). The BQ is a validated self-administered questionnaire, developed by Netzer [20], with a validated Arabic version [25] and was designed to be completed by patients during visits to their physician. Patients complete a 10-item questionnaire with questions assessing three domains: Snoring, waketime sleepiness or fatigue and hypertension. This questionnaire is generally completed in less than 30 s, with scoring taking 20 s.

Epworth Sleepiness Scale (ESS). The ESS is a validated self-administered instrument which aims to assess daytime sleepiness or sleep propensity in adults. The instrument, developed by Johns [23], with a validated Arabic version [26], contains eight items assessing the likelihood of dozing in eight specific circumstances on a scale: 0 = would never doze; to 3 = high chance of dozing. The circumstances are: Sitting and reading, watching TV, sitting inactive in a public place, as a passenger in a car for an hour without a break, lying down to rest in the afternoon when circumstances permit, sitting and talking to someone, sitting quietly after lunch without alcohol, in a car while stopped for a few minutes in traffic. This questionnaire is generally completed in 40 s, with scoring taking 10 s. 

Patients were also asked to report their age, sex, height and weight (for the BMI calculation). Once completed, the instrument was scored by the physician using the scoring guide. The information regarding a patient’s diagnosis of other medical conditions was gathered from their electronic medical records at the time of appointment. Diagnosis of these conditions was based on electronic ICD 9–10 coding of these conditions by the physicians. The medical conditions were classified as follows: Hypertension was blood pressure greater than 140/90 [27]; diabetes mellitus was fasting blood glucose greater than 7 and HbA1c greater than 6.4; hyperlipidemia was total lipid levels greater than 5; hypothyroidism was thyroid stimulating hormone (TSH) greater than 4.6 and T4 less than 5.

Code numbers were allocated to patients to protect identification. All aspects of the study were granted formal approval from the Institutional Review Board (protocol number IRB 2014-06) of King Faisal Specialist Hospital and Research Centre in Jeddah. Completion of the questionnaires was considered as implied informed consent from the participants.

Tabular analyses of the BQ and the ESS, and a measure that combined participants who were classified as high risk for sleep apnea on both the BQ and the ESS (referred to as the combined measure), were conducted to show those at high risk of sleep apnea by a series of respondent characteristics. Chi-square tests of independence were conducted on the categorical variables examining the statistical significance of associations between the proportions of those who scored as high risk versus low risk for sleep apnea on the BQ, the ESS and the combined measure by the respondent characteristics. Potential predictors of those scoring as high risk of sleep apnea and daytime sleepiness on the BQ, the ESS and the combined measure were assessed in a multivariable logistic regression model. The modelling procedure used forward and backward selection removing potential predictor variables *p* > 0.10, keeping the demographic variables (age, sex and nationality) in the models. Data were analyzed using SAS/STAT^®^ software version 9.4 (SAS Institute, Cary, NC, USA) [28]. The level of statistical significance for all analyses was set at *p* < 0.05. A post hoc power calculation showed that at 80% power and α = 0.05 the sample size of *n* = 319 exceeded the minimum requirements to detect differences between the predictor variables and the combined BQ and ESS measure.

## 3. Results

Overall 319 (147 male and 172 female) respondents gave their completed questionnaire to the attending physician. Table 1 shows the percentages of respondents in the whole sample (all respondents), those who scored as high risk for sleep apnea on the BQ, those who scored as high risk for sleep apnea on the ESS, and those who scored as high risk for sleep apnea on the combined measure, by: Sex, nationality (Saudi/non-Saudi), age group, and smoking status, and by presence of medical conditions: Obesity, hypertension, diabetes, hypothyroidism, and hyperlipidemia. 

For the whole sample, respondent distribution over the categories in Table 1 showed slightly more female than male research respondents (females = 53.9%; males = 46.1%) and more than 43% of respondents classified as being obese. Approximately one-third of respondents had been previously diagnosed with hypertension, diabetes, and hyperlipidemia; 16% had been diagnosed with hypothyroidism. 

After scoring, 95 out of 319 respondents (30%) who completed the BQ were classified as high risk for sleep apnea [20]. Statistically significantly different proportions (*p* < 0.05) were found for respondents by age group, obesity classification, hypertension, diabetes and hyperlipidemia. More specifically, Table 1 shows that the majority of respondents who were classified as high risk on the BQ was in the age category 46–60 years and had higher rates of obesity, hypertension and hyperlipidemia. 

Table 1 also shows that for the ESS [29], 102 out of 319 respondents (32%) were classified as being high risk for sleep apnea based on reports of excessive daytime sleepiness. Upon classification, the respondent profile for those scoring as high risk on the ESS was very similar to the respondent profile for the whole sample. Chi-square tests of independence for those classified as high risk for sleep apnea versus those who were not, by respondent characteristics, were not statistically significant (*p* > 0.05). 

The classified BQ and ESS variables were cross tabulated (see Methods) showing that 41 (12.9%) respondents were classified as high risk for sleep apnea on the combined measure. The Chi-square test of independence was statistically significant (*p* < 0.05) for three respondent characteristics: Obesity, hypertension and hyperlipidemia. Specifically, of the respondents classified as high risk for sleep apnea on the combined BQ and ESS measure, 32 (38.1%) were obese, 26 (31.7%) had hypertension, and 25 (33.8%) had hyperlipidemia (see Table 1).

Potential predictors of sleep apnea—hypertension, obesity, diabetes, hyperlipidemia, hypothyroidism and smoking—and selected demographic variables of age, sex and nationality were entered into three logistic regression models predicting a high risk for sleep apnea on the BQ, the ESS and the combined measure of both the BQ and the ESS. 

Table 2 shows the results of a logistic regression analysis examining the predictors of high risk for sleep apnea using the BQ. After controlling for age, sex and nationality, a diagnosis of hypertension and obesity were statistically significant predictors of being classified as at high risk of sleep apnea when using the BQ.

Table 3 shows the results of a logistic regression analysis examining predictors of high risk for sleep apnea using the ESS. After controlling for age, sex and nationality, no statistically significant predictors were found for the ESS. 

Table 4 shows the results of a logistic regression analysis examining predictors of high risk for sleep apnea using the combined measure. The table shows that those with hypertension, compared to those with no hypertension had 4 times the odds of being classified as high risk for sleep apnea; those classified as obese (compared to those not obese) had 5.7 times the odds of being classified as high risk for sleep apnea. Age, sex and nationality were not statistically significantly associated with the combined measure. No interactions were found between age, sex or nationality and the potential predictor variables, suggesting that being classified as high risk for sleep apnea did not vary by the selected demographic variables. In addition, age by sex interactions were not statistically significant in models predicting either the BQ or the ESS. Due to the non-predictive nature of the demographic variables, these variables were removed from the analysis, which did not change the results substantially and the conclusions of the analysis remained the same.

## 4. Discussion

This study reports the prevalence of being at high risk of sleep apnea over a three-month period among patients at a medical clinic in Jeddah City. High risk of sleep apnea was established using two screening measures, the BQ [20] and the ESS [23]. Overall, the BQ showed a prevalence of high risk of sleep apnea of 29.8%, while the ESS showed a prevalence of a disorder related to sleep apnea or excessive sleepiness of 32.0%. Using both the BQ and the ESS (the combined measure) the results showed that 12.9% of patients were classified as being at high risk of sleep apnea. This was slightly higher than the 8.5% sleep apnea prevalence reported in a study on Saudi adults by Wali, Abalkhail, and Krayemet [30]. In that study, despite ethnic differences, the results were similar to the sleep apnea prevalence in Western countries, China, India and Korea.

The results showed that high risk of sleep apnea was statistically significantly associated with the respondent characteristics of obesity and hypertension. High risk of sleep apnea in the combined measure was not statistically significantly predicted by diabetes, hypothyroidism or hyperlipidemia. In addition, no age or sex associations were found for those classified as high risk on a combined measure of the BQ and the ESS. This is consistent with some of the findings of Wali et al. [30], which concluded that age, male gender, obesity and hypertension are the most significant risk factors for sleep apnea. A study by Shaw, Punjabi and Wilding et al. [31] concluded a high prevalence of sleep apnea in patients with type 2 diabetes and vice versa.

This is the first study to report patient assessment of sleep apnea in Saudi Arabia using both the BQ and the ESS. Two previous assessments of sleep apnea among Saudi Arabian adults was conducted nearly 10 years ago by BaHammam et al. [17,18] and reported that 33.3% of men and 39% of women were considered as high risk for sleep apnea when using the BQ. When using only the BQ, the present study showed that 27.9% of men and 31.4% of women were classified as high risk for sleep apnea, which were lower rates than those previously reported by BaHammam et al. [17,18]. If only respondents aged 46–60 years were examined (the age range for the BaHammam et al. [17,18] studies), the BQ showed a rate of 40% of respondents being classified as high risk, a rate that is similar to the women in the BaHammam et al. study.

Previous research has documented that those who are middle-aged or older are more likely to exhibit sleep apnea symptoms [2]. The findings of the current study were not consistent with this, showing that age was only statistically significantly associated in the bivariate Table 1 analyses of the BQ, and did not persist in the logistic regression analyses. Given the paucity of sleep apnea research across a wide age range (18–80 years) in Saudi Arabia, there is little contemporaneous data on this issue with which to compare these findings.

Overall, it can be seen that the use of the ESS did not yield many statistically significant predictors of sleep apnea, despite being a validated measure [23,26]. These results may be taken to imply that the ESS did not discriminate those with sleep apnea as effectively as the BQ. In addition, the Nagelkerke R-square test of the logistic regression models, was highest for the BQ (0.30), and lowest for the ESS (0.04), with the combined measure being 0.20. However, when assessing the performance of the two questionnaires, it is important to be aware of the purpose of the questionnaire. The ESS was developed primarily to assess daytime sleepiness. Predictors of daytime sleepiness in the ESS are likely to be different to those of the BQ, which was primarily developed to detect snoring. Therefore, factors associated with snoring in the BQ, such as clinical health problems, are likely to differ from factors associated with daytime sleepiness, such as accidents and cognitive problems. The current study did not assess accidents or cognitive problems, so it is recommended that physicians use both questionnaires in their practice. To do so will only take about 2 minutes of the appointment time and is a cost-effective approach to identifying patients at high risk of sleep apnea prior to potentially invasive and expensive testing. 

One of the main strengths of this study is that it is the first study to report the patient assessment of sleep apnea in Saudi Arabia using both the BQ and the ESS, both validated measures of sleep apnea [20,32] and excessive daytime sleepiness [23,26]. The benefits of screening patients in this manner may prevent unnecessary referrals for sleep apnea investigations making it worth the short time to complete and score both questionnaires in an ambulatory care clinic setting. 

However, some limitations need to be addressed. This was a cross-sectional study of a sample of patients at a medical clinic and it is unclear whether those classified as being at high risk of sleep apnea would also have been classified as having sleep apnea on overnight polysomnography. Hence, these questionnaires may be useful tools to assess risk of sleep apnea while accepting that polysomnography is considered as the gold standard for the diagnosis of sleep apnea. Patients presenting with obesity and hypertension may be at high risk of sleep apnea and excessive daytime sleepiness and should be a priority for screening. Further, it should be acknowledged that obesity and hypertension may be both causes and consequences of sleep apnea; the study design prevented any analysis of causal pathways. 

Future research should conduct overnight polysomnography on patients to determine sleep apnea status. In addition, research on the predictors of measures such as the ESS should include potential predictors of sleepiness such as workplace accidents and cognitive problems.

## 5. Conclusions

In conclusion, the screening revealed that when using both the BQ and the ESS, approximately 3/10 patients were found to be at high risk of sleep apnea. Given the potentially adverse health and social consequences of sleep apnea and excessive sleepiness, it is important that physicians can recognize sleep apnea in their patients to facilitate timely clinical help to combat this under-diagnosed but remediable ailment.

## Figures and Tables

**Table 1 ijerph-16-00459-t001:** Descriptive statistics of study respondents assessed as at high risk of sleep apnea.

Respondent Characteristic	All Respondents *n* = 319, %	Berlin Questionnaire (High Risk) *n* = 95, %	*p*	Epworth Sleepiness Scale (High Risk) *n* = 102, %	*p*	Epworth x Berlin (High Risk) *n* = 41, %	*p*
Sex							
Male	46.1	43.2		46.1		45.7	
Female	53.9	56.8		53.9		54.3	
			0.495		0.999		0.710
Nationality							
Saudi	76.2	76.8		76.5		76.3	
Non-Saudi	23.8	23.2		23.5		24.4	
			0.856		0.932		0.927
Age Group							
18–30 years	26.0	12.6		34.3		27.0	
31–45 years	28.2	22.1		27.5		29.5	
46–60 years	27.3	40.0		25.5		25.5	
61–80 years	18.5	25.3		12.8		18.0	
			<0.001		0.077		0.194
Smoking status							
Smoker	19.6	15.4		22.5		20.1	
Non-smoker	80.4	84.6		77.6		79.9	
			0.234		0.383		0.533
Obese (BMI > 30 kg/m^2^)							
Yes	43.3	73.7		43.1		38.1	
No	56.7	26.3		56.9		61.9	
			<0.001		0.976		<0.001
Hypertension							
Yes	35.7	60.0		35.3		31.7	
No	64.3	40.0		64.7		68.4	
			<0.001		0.910		<0.001
Diabetes							
Yes	28.2	42.1		25.5		26.6	
No	71.8	57.9		74.5		73.4	
			<0.001		0.459		0.099
Hypothyroidism							
Yes	16.0	22.1		14.7		15.1	
No	84.0	77.9		85.3		84.9	
			0.052		0.669		0.264
Hyperlipidemia							
Yes	37.3	53.7		35.3		33.8	
No	62.7	46.3		64.7		66.2	
			<0.001		0.611		<0.001

**Table 2 ijerph-16-00459-t002:** Odds ratios and 95% confidence intervals for predictors of high risk for sleep apnea using the Berlin questionnaire.

Measure	OR	95% CI	*p*
Demographic variables			
Age			0.378
18–30 years (reference)	1		
31–45 years	0.87	0.36–2.09	
46–60 years	1.67	0.69–4.04	
61–80 years	1.19	0.42–3.38	
Sex			0.208
Male	0.69	0.39–1.23	
Female (reference)	1		
Nationality			0.266
Saudi	0.67	0.34–1.35	
Non–Saudi (reference)	1		
Participant diagnosed with hypertension			<0.001
Yes	3.49	1.78–6.83	
No (reference)	1		
Participant obese			<0.001
Yes	5.78	3.23–10.34	
No (reference)	1		

**Table 3 ijerph-16-00459-t003:** Odds ratios and 95% confidence intervals for predictors of high risk for sleep apnea using the Epworth Sleepiness Scale (ESS).

Measure	OR	95% CI	*p*
Demographic variables			
Age			0.073
18–30 years (reference)	1		
31–45 years	0.63	0.34–1.18	
46–60 years	0.58	0.31–1.09	
61–80 years	0.37	0.17–0.80	
Sex			0.957
Male	1.01	0.62–1.65	
Female (reference)	1		
Nationality			0.615
Saudi	1.16	0.65–2.08	
Non-Saudi (reference)	1		

**Table 4 ijerph-16-00459-t004:** Odds ratios and 95% confidence intervals for predictors of high risk for sleep apnea using the combined Berlin Questionnaire and Epworth Sleepiness Scale measures.

Measure	OR	95% CI	*p*
Demographic variables			
Age			0.337
18–30 years (reference)	1		
31–45 years	0.38	0.12–1.23	
46–60 years	0.57	0.18–1.81	
61–80 years	0.36	0.09–1.40	
Sex			0.911
Male	0.96	0.46–1.99	
Female (reference)	1		
Nationality			0.366
Saudi	0.67	0.27–1.61	
Non–Saudi (reference)	1		
Participant diagnosed with hypertension			0.003
Yes	3.99	1.60–9.97	
No (reference)	1		
Participant obese			<0.001
Yes	5.73	2.50–13.10	
No (reference)	1

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
