# Peer review of "Screening for High Risk of Sleep Apnea in an Ambulatory Care Setting in Saudi Arabia"

_ijerph, 2019, doi:10.3390/ijerph16030459_

Round 1

Reviewer 1 Report

General

The topic of this paper is scientifically important. The aim was to evaluate the screening capability of two questionnaires to identify sleep apnea in Saudi Arabi. The authors find that both questionnaires (BQ and ESS) show promise, particularly when combined. This is publishable but only after a few considerations. There are no analyses performed to suggest that the combined assessment is better to assess prevalence than each questionnaire separately. Because of this, I would like to see logistic models for each questionnaire separately. Also the discussion section is lacking any perspective of the ambulatory care piece. Please find my detailed comments below:

Title

-Good title

Abstract

-Well-written abstract, no comments.

Introduction

-Well written introduction, no comments.

Methods

-Can you provide the citations that test internal validity of the questionnaires that were in Arabic?

-Hypertension guidelines have been updated to 130/80; I suggest updating this variable. It should not change the results drastically.

Whelton, P. K., Carey, R. M., Aronow, W. S., Casey, D. E., Collins, K. J., Himmelfarb, C. D., ... & MacLaughlin, E. J. (2018). 2017 ACC/AHA/AAPA/ABC/ACPM/AGS/APhA/ASH/ASPC/NMA/PCNA guideline for the prevention, detection, evaluation, and management of high blood pressure in adults: a report of the American College of Cardiology/American Heart Association Task Force on Clinical Practice Guidelines. Journal of the American College of Cardiology, 71(19), e127-e248.

-It is not clear to me how the responses from BQ and ESS were combined. I am assuming it is the proportion of participants who were detected to be high risk via both scales? This should be explicit in the methods section.

Results

-Replace “gender” with “sex”.

-Have you looked at the odds of high risk sleep apnea with the questionnaires separately? I would like to see another table (or figure) that shows which questionnaire is driving the predictors found in the combined sleep apnea variable. From Table 1, the combined subgroup looks a lot like the ESS+ group. As a side note, I think the “positive” label would be better reworded as “high risk” or “sleep apnea” or some other alternative.

Discussion

-As a general comment, I would like to know if the authors can determine if primary-care physician should use both questionnaires to determine risk or can she/he use either one? You only present one model using combined but perhaps, even though the questionnaires separately capture higher percentages, they might do the trick for primary care. It may be better for high specificity and have high false negatives for surveillance purposes. If the authors feel another way, there should be discussion nonetheless.

-I think the prevalence of sleep apnea should be presented in all three ways. There are no analyses performed that show that the combined assessment is better than either questionnaire respectively.

-Can the authors add a paragraph or some sentences contrasting the prevalence in Saudi Arabia with sleep apnea prevalence with other countries? Also reflect on similarities or differences found.

-Add strengths of the study to limitations paragraph.

Author Response

Dear Reviewer,

The authors would like to thank the reviewers for their consideration and comments. We would like to address their suggestions as follows:

Reviewer 1.

Methods

1. Can you provide the citations that test internal validity of the questionnaires that were in Arabic?

Reply. We have now inserted the following citations: validation of Arabic version of Berlin questionnaires reference (Saleh et al 2011) (see line 90); validation of Arabic version of Epsworth Sleepiness Scale reference (Ahmed et al 2014) (see line 96).

2. Hypertension guidelines have been updated to 130/80; I suggest updating this variable. It should not change the results drastically. Whelton, P. K., Carey, R. M., Aronow, W. S., Casey, D. E., Collins, K. J., Himmelfarb, C. D., ... & MacLaughlin, E. J. (2018). 2017 ACC/AHA/AAPA/ABC/ACPM/AGS/APhA/ASH/ASPC/NMA/PCNA guideline for the prevention, detection, evaluation, and management of high blood pressure in adults: a report of the American College of Cardiology/American Heart Association Task Force on Clinical Practice Guidelines. Journal of the American College of Cardiology, 71(19), e127-e248

Reply. In this study we defined hypertension as systolic BP is 140/90. This decision was based on the UK National Institute of Clinical Excellence (NICE) and British Hypertension Society guidelines   https://www.nice.org.uk/guidance/CG127/chapter/1-Guidance#diagnosing-hypertension-2

Therefore, we decline this author suggestion, but we have included a reference for this guideline on line 108.

3. It is not clear to me how the responses from BQ and ESS were combined. I am assuming it is the proportion of participants who were detected to be high risk via both scales? This should be explicit in the methods section.

Reply. The reviewer is correct that participants, who were classified as high risk for sleep apnea on both the Berlin Questionnaire and the Epsworth Sleepiness Scale, were selected for the combined measure. We have added a sentence on how the measures were combined (see line 116).

 Results

4. Replace “gender” with “sex”.

Reply. We have replaced the word gender with sex throughout the manuscript.

5. Have you looked at the odds of high risk sleep apnea with the questionnaires separately? I would like to see another table (or figure) that shows which questionnaire is driving the predictors found in the combined sleep apnea variable. From Table 1, the combined subgroup looks a lot like the ESS+ group. As a side note, I think the “positive” label would be better reworded as “high risk” or “sleep apnea” or some other alternative.

Reply. The data has been reanalysed using logistic regression and two tables have been produced showing the predictors of sleep apnea using the Berlin Questionniare (Table 2) and the Epsworth Sleepiness Scale (Table 3) are now also included in the paper (see also line 118).

However, it is difficult to determine the drivers of the predictors of sleep apnea using the Berlin and the Epsworth questionnaires in the current study. This is because we would need to know definitively who did/did not have sleep apnea using a procedure such as overnight polysomnography. While both questionnaires have been validated, they only indicate high risk for sleep apnea; assessment of sensitivity and specificity of these questionnaires in this population was outside the scope of this study. 

We have also changed the wording from positive to “high risk for sleep apnea” throughout the manuscript.

 Discussion

6. As a general comment, I would like to know if the authors can determine if primary-care physician should use both questionnaires to determine risk or can she/he use either one? You only present one model using combined but perhaps, even though the questionnaires separately capture higher percentages, they might do the trick for primary care. It may be better for high specificity and have high false negatives for surveillance purposes. If the authors feel another way, there should be discussion nonetheless.

Reply. We believe that primary care physicians would be best to use both questionnaires for the following reasons. While the Epsworth Sleepiness Scale does not have any specific predictors associated with high risk for sleep apnea, the use of it with the Berlin Questionnaire (which does have predictors of hypertension and obesity), will most likely reduce the detection of false positive results. The administration and scoring of both questionnaires is very short (less than 2 minutes), and should not be too onerous on the physician or the patient.

In addition, while the Berlin Questionnaire and Epsworth Sleepiness Scale have differing prediction profiles, and while the Berlin Questionnaire is better explained by the selected predictors than the Epsworth Sleepiness Scale, this does not necessarily mean that only the Berlin Questionnaire should be used. This is because problems of sleepiness that could be indicators of sleep apnea such as car accidents, workplace accidents, reduced concentration, memory, should also be considered rather than just clinical health problems. This issue will be discussed in the Discussion for future research see line 230 and 259.

7. I think the prevalence of sleep apnea should be presented in all three ways. There are no analyses performed that show that the combined assessment is better than either questionnaire respectively.

Reply. We have reanalysed the data and now include two additional analyses (see reviewer point 5) showing predictors of sleep apnea on the Berlin Questionnaire and the Epsworth Sleepiness Scale, which are shown in Table 2 and Table 3. From this information it was possible to perform a Nagelkerke R-square test. The results showed that the Berlin Questionnaire did marginally out perform the combined measure (0.3 versus 0.2). The Epsworth Sleeiness Scale R-square was 0.04 (see line 233). However, as explained in reviewer point 6 if is possible that the sleep apnea on the Epsworth Sleepiness Scale was not well predicated using the chosen clinical health measures rather than sleepiness measures (accidents/cognitive problems). This issue will be discussed in the Discussion see line 230 and the section on future research see line 259.

8. Can the authors add a paragraph or some sentences contrasting the prevalence in Saudi Arabia with sleep apnea prevalence with other countries? Also reflect on similarities or differences found.

Reply. We have added some information comparing prevalence in Saudi Arabia to other countries (see line 205).

9. Add strengths of the study to limitations paragraph.

Reply. We have now added strengths in paragraph above the limitations paragraph (see line 244).

Reviewer 2 Report

Well written introduction clearly describing the relevant literature, and the study’s aim.  Please emphasise that while questionnaires may be useful in identifying persons at risk of sleep apnea the disorder is diagnosed by overnight polysomnography.   

Can the authors suggest why age, which is positively related with apnea risk in the literature, did not predict a high risk for sleep apnea on the combined measure of both the BQ and the ESS?  

How long does it take to complete the BQ and ESS?  Is it practical for general practitioners/family doctors to use both questionnaires during a consultation (time constraints, ability to score and interpret results, etc?).

In this study high risk of sleep apnea was predicted by obesity and hypertension.  I appreciate this was not the study’s focus but please acknowledge that both disorders may be causes as well as consequences of sleep apnea.   

Author Response

Dear Reviewer, 

The authors would like to thank the reviewers for their consideration and comments. We would like to address their suggestions as follows:

10. Well written introduction clearly describing the relevant literature, and the study’s aim.  Please emphasise that while questionnaires may be useful in identifying persons at risk of sleep apnea the disorder is diagnosed by overnight polysomnography.   

Reply. We have added a line to the manuscript to address this (see line 252).

11. Can the authors suggest why age, which is positively related with apnea risk in the literature, did not predict a high risk for sleep apnea on the combined measure of both the BQ and the ESS?  

Reply. In the Table 1 bivariate analysis, age was associated with being at high risk for sleep apnea on the Berlin Questionnaire; and on the Epsworth Sleepiness Scale the association tended towards statistical significance. The association did not persist on the combined measure. Upon examination, the age patterns are very different, and combine to produce a rather flat distribution – hence the lack of statistical significance. In the logistic regression models, again age was not a statistically significant predictor of sleep apnea in any of the questionnaires or combined measure.

It is unclear what the reason is and to do so would involve speculation. Potentially the previous literature has been derived from other populations of differing ethnicities and age ranges; differing sleep apnea assessment tools were used; and differing study designs may have been employed. These studies may not be comparable to the current study for these reasons and as so few studies have been conducted in Saudi Arabia, there is little contemporaneous data on this population with which to compare.

This issue has been addressed in the Discussion, see line 224.

12. How long does it take to complete the BQ and ESS?  Is it practical for general practitioners/family doctors to use both questionnaires during a consultation (time constraints, ability to score and interpret results, etc?).

Reply. We have added following details page line 92 (Berlin Questionnaire) and 101 (Epsworth Sleepiness Scale). The time taken to complete and score questionnaires is as follows: Berlin = completing less than 30 seconds and scoring 20 seconds = 50 seconds. Epsworth Sleepiness Scale = completing 40 seconds and scoring 10 seconds = 50 seconds. Total time to complete and score both questionnaires = 1min 40 seconds approximately, may take up to 2 minutes in patients who need to think a bit about their answers. This is feasible and practical in addition to the time and money saved in unnecessarily referring patients for further sleep apnoea investigations.

13. In this study high risk of sleep apnea was predicted by obesity and hypertension.  I appreciate this was not the study’s focus but please acknowledge that both disorders may be causes as well as consequences of sleep apnea

Reply. The authors agree that obesity and hypertension may be both causes and consequences of sleep apnea. We have added this into the Discussion on page limitations (see line 255).

Reviewer 3 Report

The paper’s aim is to propose the use of Berlin Questionnaire and the Epworth Sleeping Scale to screen patients for risk of sleep apnea. The method has been used in a family medicine clinic in Saudi Arabia, for a limited amount of time, over three years ago.

I don’t recommend the publication because it is not clear the validity of this method to screen patients for sleep apnea. The results and the statistical approach need to be presented in a better and clearer way. The given explanation on why no sample size calculation was seemed necessary (rows 84-85) is not at all clear to me.

The results are not clearly exposed, and also the statistic is not clearly presented. The table 1 is terrible. It is difficult to read it. It could be useful to provide horizontal lines for guiding the eye. One could even think of taking the transpose of the table, maybe it would be clearer.

It is well known that hypertension, diabetes, obesity, and smoking, are all major risk factors for sleep apnea. The application together of Berlin Questionnaire and the Epworth Sleeping Scale to screen patients attending a family medicine clinic in Saudi Arabia for risk of sleep apnea doesn’t seem to improve the predictive capacity of these tests to detect the risk of sleep apnea. In particular, the fact that the ESS would increase the statistical significance of detection of sleep apnea risk is not clear at all, at least based on this data.

There are a lot of English imprecisions, e.g:

Line 33 some medications..certain

69 that time prevalence…in this time the prevalence of

82 of the principal..owned by the

107-109 of more than…greater

Author Response

Dear Reviewer, 

The authors would like to thank the reviewers for their consideration and comments. We would like to address their suggestions as follows:

14. I don’t recommend the publication because it is not clear the validity of this method to screen patients for sleep apnea. The results and the statistical approach need to be presented in a better and clearer way. The given explanation on why no sample size calculation was seemed necessary (rows 84-85) is not at all clear to me.

Reply. The Berlin Questionnaire and the Epsworth Sleepiness Scale have been previously validated, in the current study the Berlin Questionnaire and the Epsworth Sleepiness Scale were used in their original format, with the addition of those who wanted to use the validated translated forms (see line 90 (Berlin Questionnaire) and 96 (Epsworth Sleepiness Scale)).

The statistical methods and tables have been revised in the light of reviewer comments.

Upon reflection, we conducted a post hoc power calculation for this study. Overall, we found that for each of the predictors in this study, we had exceeded the minimum suggested sample sizes at 80% and α=0.05 to detect differences between the predictor variables and the combined measure of the Berlin Questionnaire and Epsworth Sleepiness Scale. We have amended our paragraph in the manuscript to reflect this (see line 127).

15. The results are not clearly exposed, and also the statistic is not clearly presented. The table 1 is terrible. It is difficult to read it. It could be useful to provide horizontal lines for guiding the eye. One could even think of taking the transpose of the table, maybe it would be clearer.

Reply. We have revised Table 1 for clarity.

16. It is well known that hypertension, diabetes, obesity, and smoking, are all major risk factors for sleep apnea. The application together of Berlin Questionnaire and the Epworth Sleeping Scale to screen patients attending a family medicine clinic in Saudi Arabia for risk of sleep apnea doesn’t seem to improve the predictive capacity of these tests to detect the risk of sleep apnea. In particular, the fact that the ESS would increase the statistical significance of detection of sleep apnea risk is not clear at all, at least based on this data.

Reply. This issue has been addressed in reviewer points 6 and 7. The authors agree that at first glance the Epsworth Sleepiness Scale does not appear to improve the detection of sleep apnea in this sample. However, we believe that as sleepiness was  not addressed in the selected predictors, the Epsworth Sleepiness Scale did not appear to aid the prediction of sleep apnea. This issue has been discussed in the Discussion see line 230 and is also included in a section on future research see line 259.

17. There are a lot of English imprecisions, e.g:

Line 33 some medications..certain

69 that time prevalence…in this time the prevalence of

82 of the principal..owned by the

107-109 of more than…greater

Reply. These have been amended.

Round 2

Reviewer 1 Report

All of my original comments have been addressed.

Reviewer 3 Report

The authors have revised the papers following suggestion and the manuscript seems better now and publishable.